# "If my husband was in the labor ward with me, my baby wouldn't have died"; experiences on birth companionship from a tertiary health facility, Tanzania

**Michael Obed Mwakyusa**[1,2]\*, **Ali Said**[1], **Shekha Selemani**[1], **Musa Kakiziba**[1], **Judica Christopher**[1,3], **Nathanael Shauri Sirili**[4], **Fadhlun M. Alwy Al-beity**[1]

1 Department of Obstetrics and Gynecology, Muhimbili University of Health and Allied Sciences, MUHAS, Dar es Salaam, Tanzania, 2 Department of Obstetrics and Gynecology, Mbeya Zonal Referral Hospital, MZRH, Mbeya, Tanzania, 3 Department of Obstetrics and Gynecology, University of Dar Es Salaam, UDSM —MCHAS, Mbeya, Tanzania, 4 Department of Development Studies, Muhimbili University of Health and Allied Sciences, MUHAS, Dar es Salaam, Tanzania

\* michaelmwakyusa@yahoo.com

**Data Availability Statement:** Data cannot be shared publicly due to containing sensitive participants' information and participants not consenting to having their interviews shared. Data

## Abstract

### Background

Despite existing policies promoting companionship, it remains uncommon in Tanzania. Pregnant women select a trusted individual to accompany them during childbirth, providing emotional, physical, and spiritual support. The World Health Organization recommends birth companionship as integral to intrapartum care for positive maternal and fetal outcomes.

### Objective

This study aimed to explore the experiences of pregnant women and healthcare providers regarding childbirth companionship at a tertiary health facility in Tanzania.

### Methods

Participants were purposefully selected for qualitative exploratory interviews. Focused group discussions were conducted with pregnant women attending antenatal clinics, and in-depth interviews were held with healthcare providers at Mbeya Zonal Referral Hospital in Tanzania. Braun and Clarke's six-step thematic analysis approach was used to analyze the data.

### Results

Three major themes emerged: *"Bonding and Learning with Spouse/Partner," "Assurance of Maternal Safety,"* and *"Fear of Blame and Breach of Confidentiality."* These themes highlight a strong desire among both pregnant women and healthcare providers for companionship during labor, particularly from male partners. Emotional support and safety were cited as key reasons. Fear, however, was identified as a major obstacle, with providers concerned

are available from the Directorate of Research and Publications at the Muhimbili University of health and allied sciences (contact via drp@muhas.ac.tz) for researchers who meet the criteria for access to confidential data.

**Funding:** The author(s) received no specific funding for this work.

**Competing interests:** The authors have declared that no competing interests exist.

**Abbreviations:** ANC, Antenatal Clinic; FGD, Focused Group Discussion; IDI, In-Depth Interview; MCHAS, Mbeya College of Health and Allied Science; MUHAS, Muhimbili University of Health and Allied Sciences; MZRH, Mbeya Zonal Referral Hospital; RMNCAH, Reproductive Maternal Newborn Child and Adolescent Health; SES, Socio-Economic Status; UDSM, University of Dar es Salaam; WHO, World Health Organization.

about potential exposure of mistakes and pregnant women fearing a violation of their privacy rights regarding health matters.

## Conclusion

Pregnant women expressed a strong desire for companionship throughout labor. However, companionship faced challenges due to staff shortages and inadequate privacy in labor wards. There is a pressing need to enhance childbirth companionship practices and policies in low-resource settings.

## Introduction

Worldwide, around 140 million births occur yearly, with most being vaginal deliveries and occurring without complications for both mothers and newborns [1]. In situations where complications arise, both the mother and newborn face a high risk of serious morbidity and mortality [2]. Over one-third of maternal deaths and a substantial proportion of pregnancy-related life-threatening conditions are attributed to the complications that arise during labor or the immediate period after it [2, 3]. In 2020, an estimated 287,000 maternal deaths occurred worldwide, with approximately 202,000 of these deaths in sub-Saharan Africa due to pregnancy and childbirth complications [4]. Low-income countries have a higher burden of maternal and perinatal deaths than high-income countries [5]. Hence, improving the quality of care during the time of labor and delivery, especially in low-income countries, has been identified as the most important strategy for reducing childbirth associated deaths [6, 7].

Birth companionship refers to support provided to a woman during labor and delivery. It may be provided by a partner, family member, friend, doula, or healthcare professional [8]. The World Health Organization (WHO) and previous research recommend birth companionship as critical in providing respectful maternity care for a positive childbirth experience and to improve the quality of maternal and newborn care in health facilities [9–12]. The WHO recommends respectful communication between healthcare professionals and women in labor, and that all women undergoing childbirth receive continuous emotional support. Despite its importance, companionship is not widely practiced in all clinical settings [12, 13].

Power, passage, passenger and psyche are some of the variables that affect the likelihood of a successful pregnancy outcome [13]. Research indicates that expectant mothers who receive unbroken support throughout their labor are more likely to give birth naturally, resulting in fewer C-sections or surgical deliveries; their labor lasts shorter; they require less intrapartum analgesia; they are happier with their childbirth experience; and their newborns are less likely to have a low five-minute Apgar score [14, 15]. Birth companions provide women with physical, emotional, and spiritual support during labor and delivery, which benefits them and improves the outcomes of childbirth.

Companionship may help to reduce staff workload and improve processes of childbirth [10]. Companions can help by attending to urgent issues and reminding staff of re-examinations or changes. They can arrange transportation for referrals if complications arise and reinforce messages and instructions to women [14]. Allowing and supporting the presence of a woman's companion of choice during childbirth potentially reduces mistreatment or abuse in a health facility [16]. Common components of support provided by childbirth companions include emotional support, informational support, comfort measures, advocacy and oral sips feed [12, 17, 18].

The implementation of companionship during childbirth is challenging due to the influence of various factors like physical environment and sociocultural norms [17]. Young women are more likely to seek companionship, and factors such as socioeconomic status (SES) and religious beliefs can also play a role. For instance, some religions, like Orthodox faiths, may disagree with the idea of companionship during childbirth. Similarly, certain tribes, such as the Yoruba in Nigeria, may also have cultural norms that discourage companionship [18]. It is crucial for healthcare providers to be aware of and respect women's preferences in this regard. Women may require assistance at any stage of labor, delivery, or recovery, or they may prefer to manage these processes independently. Healthcare providers should prioritize understanding and accommodating these preferences [14].

Moreover, it is essential for healthcare providers to identify women's chosen companions accurately. The person who accompanies a woman to a health institution may not necessarily be her preferred companion. Antenatal care provides an opportunity to educate women and their companions about the importance of emotional support and the necessity of having their preferred companions with them at the delivery facility [19].

The World Health Organization has been actively working to reduce maternal mortality and morbidity by improving the availability and accessibility of facility-based birthing services. This effort has led to an increase in the number of babies born in hospitals worldwide [20]. With this shift, there is a growing emphasis on enhancing the quality of care provided at facilities. Improving women's care experiences is a key component of enhancing maternal and neonatal health outcomes [21]. Most studies indicate that both women and healthcare providers advocate for continuous childbirth companionship to improve maternal and fetal outcomes [20, 22]. In the general population, women often prefer female companions over male partners, regardless of their level of education or beliefs [7, 19, 20, 23]. However, male partners who do accompany women during childbirth often report positive impacts on their relationship with the woman and the new baby. Despite this, some men may feel anxious or scared witnessing their partners in pain [22]. False beliefs and inadequate infrastructure in delivery rooms have been shown to discourage women from having their preferred companions during childbirth [24, 25].

The World Health Organization has issued two guidelines recommending the provision of continuous companionship during childbirth [9, 14]. While Tanzania's National Guidelines on Gender and Respectful Care Mainstreaming and Integration across Reproductive Maternal Newborn Child and Adolescent Health (RMNCAH) services emphasize the presence of a companion during childbirth, strictly stipulating that the companion must be female [25]. Despite global evidence suggesting that birth companionship can improve maternal outcomes, including reduced labor duration, decreased cesarean rates, and enhanced maternal satisfaction, there is limited understanding of how these benefits translate in the Tanzanian context. Previous study in the region have not adequately explored the desires, preferences, and perceived barriers to childbirth companionship from the perspectives of both pregnant women and healthcare providers [26]. This study seeks to address this gap by providing a detailed examination of these factors at a major healthcare facility in southern Tanzania.

## Materials and methods

### Study design and setting

An exploratory qualitative study was conducted to understand the experiences of pregnant women and healthcare providers regarding childbirth companionship at Mbeya zonal referral hospital (MZRH) in southern Tanzania. This approach was chosen for its ability to capture in-depth insights into personal experiences, beliefs, and practices, especially in a context with

limited prior research [27]. Qualitative methods allow for the exploration of complex social and cultural factors that quantitative methods might overlook. Therefore, this study aimed to explore the desire for, preferred forms of, and perceived barriers to childbirth companionship among pregnant women and healthcare providers in Tanzania.

The study was set in Mbeya Region, a diverse area in southern Tanzania. MZRH serves as a major referral center for the southern highland zone, which includes the regions of Ruvuma, Njombe, Iringa, Mbeya, Songwe, and Rukwa. The hospital also receives referrals from Katavi in the Western Zone. The total population served by these regions is approximately 11.4 million people [28]. This extensive catchment area and diverse patient population make MZRH an ideal setting for examining the dynamics of childbirth companionship. The region includes both urban and rural areas, with significant variability in income levels, education, and access to healthcare services.

The hospital handles approximately 450 deliveries per month, with about 40% of these being vaginal births. The maternity ward accommodates a significant number of women, underscoring the hospital's crucial role in regional healthcare. The hospital has a policy that allows every pregnant woman to have a companion of her choice during labor, though this policy is not consistently implemented. Ensuring consistent implementation of this policy is important as childbirth companionship benefits maternal outcomes.

## Participants and recruitment

Purposive and chain referral sampling strategies were used to recruit healthcare providers based on their years of experience [27]. The researchers requested the head of the Department of Obstetrics and Gynecology to provide a list of nurses in the labor ward and Antenatal Care (ANC) clinic who are directly involved in the care of women during ANC visits, labor, and delivery. The researchers categorized the participants into two groups: nurses with less experience, who had worked for more than 6 months but less than 4 years prior to the study, and those with high experience, who had worked in the labor ward or ANC clinic for more than 4 years. The inclusion of nurses with different years of experience aimed to uncover varying perspectives of healthcare providers towards childbirth companionship, given the previous high implementation of the practice and its recent cessation. In each group from the labor ward and ANC clinic, the researchers approached the first participant after being introduced by the head of the department and explaining the purpose of their visit. Each identified nurse then referred the researchers to another member with the same level of work experience. We obtained a list of pregnant women attending the ANC clinic from the clinic supervisor and recruited 7 to 10 participants for each focus group discussion from those who agreed to participate after being informed about the study's purpose. A total of five focus group discussions were conducted with pregnant women. We determined that saturation was reached for both in-depth interviews and focus group discussions when no new themes, insights, or information emerged from the data. This occurred after multiple rounds of each method, during which participant responses began to repeat and no additional data relevant to our research objectives were identified.

## Data collection

The principal researcher and one trained research assistant conducted in-depth interviews (IDIs) with healthcare providers and focused group discussions (FGDs) with pregnant women attending the antenatal clinic. Interviews were conducted in a private room to ensure confidentiality and create a comfortable environment for participants to freely share their experiences and perspectives. The purpose of the study was explained to each study participant,

including their rights to withdraw and the principles of confidentiality. Written informed consent was requested from each participant before the interview, for audio recording and for publishing the information collected. With permission from respondents, a digital voice recorder was used to capture audio information. The interview guide for the in-depth interviews and focus groups was developed collaboratively within the research team. The guide was designed to ensure comprehensive coverage of key topics related to childbirth companionship, drawing on both the team's expertise and relevant literature. All interviews were conducted in Swahili, the native language of both participants and researchers.

Five focus group discussions were conducted using a semi-structured interview guide. The aim was to obtain as much information as possible, given participants' varying levels of experience with labor and childbirth. The interviews began with questions about participants' demographic information, followed by more general questions with probes regarding desires, preferred companions, and barriers to childbirth companionship. The data analysis was conducted alongside data collection. After each focus group, the research team examined the transcripts to identify recurring themes and patterns. By the fifth discussion, participants were consistently mentioning the same themes, indicating that saturation had been achieved. Interviews lasted between 30 and 75 minutes.

In-depth interviews were conducted with healthcare personnel at Mbeya Zonal Referral Hospital to investigate perceived barriers to childbirth companionship. A semi-structured interview guide with open-ended questions and probes was used to explore relevant issues. Each interview was scheduled at a convenient time for the participant. Due to staff shortages and tight timetables, individual interviews were chosen over focus groups to accommodate the healthcare providers' schedules. Data collection continued until data saturation was reached after the tenth interview. Interviews lasted between 25 to 55 minutes.

## Data processing and analysis

The analysis was conducted simultaneously with data collection, commencing after the initial three interviews were completed. Audio recordings of the interviews were transcribed verbatim and subsequently translated into English. Thematic analysis, following the framework of Braun and Clarke, was employed, comprising six distinct phases. The interviews were transcribed verbatim in Swahili to preserve the authenticity of the data, with transcriptions carried out by the principal researcher and researchers experienced in qualitative research. The translated transcripts were then reviewed by the principal researcher. Two researchers independently conducted the analysis (inductive coding), which involved reading and re-reading the complete transcripts to become familiar with the data and context. Codes were developed based on the transcripts, and the initial list of codes was discussed among the authors to finalize them. Similar codes were grouped to form sub-themes, which were then reviewed for similarities and aligned with participants' experiences. The authors then deliberated and agreed upon the three main themes, supported by concise quotes selected from the transcripts.

## Methodological considerations

To establish trustworthiness in a qualitative study, it is essential that the findings are deemed credible [29]. In this study, we maintained adherence to five fundamental criteria outlined by Lincoln and Guba [30]: credibility, dependability, confirmability, transferability, and authenticity.

Credibility was ensured by using purposive sampling to select participants with diverse demographics, including age, socioeconomic status, and childbirth experience. We conducted in-depth interviews and focus group discussions, incorporating direct participant quotations to provide authentic insights and enhance the trustworthiness of our findings.

Dependability was achieved by conducting interviews and discussions with both the principal researcher and a research assistant, ensuring thorough information extraction. Open-ended questions and a predetermined line of questioning were used to ensure consistency.

Confirmability was maintained by providing detailed descriptions of the study settings, participant selection, data collection, and analysis methods, allowing for replication by other researchers.

Transferability was ensured by presenting comprehensive information about the study settings, participant selection, data collection, and analysis methods, enabling readers to assess the applicability of the findings to other contexts.

Authenticity was achieved by accurately representing participants' feelings and emotions, supported by direct quotations in the data analysis.

The study engaged maternity healthcare providers and pregnant mothers with experience at the facility, offering a detailed perspective on childbirth companionship and identifying areas for improvement.

The study's limitations include reliance on self-reported data, which may introduce response biases, and a limited sample size and setting, affecting the generalizability of findings on the desire, preferences, and barriers to childbirth companionship.

## Results

In the focus group discussions with pregnant women, most participants were in the 20–29 age group, followed by the 30–39 age group. All participants were female, and the majority had low parity (less than 3 previous pregnancies), was married, had completed primary education, and was not employed.

In the in-depth interviews with doctors and nurses, most participants were female, with a few males. Most were multiparous, married, and had varying education levels, with some having completed college or university.

From both focus group discussions and in-depth interviews, three main themes emerged from the analysis: *(1) Learning and bonding with the spouse/partner, (2) Assurance of safety to the mother and (3) Fear of blame and breach of confidentiality*. Each main theme included 2–4 sub-themes.

### Learning and bonding with the spouse/partner

Participants highlighted that childbirth companionship provided an opportunity for spouses or partners to learn about the childbirth process. This learning experience often resulted in a stronger bond between the couple. Pregnant women appreciated having their partners present during labor and delivery, as it helped them feel supported and reassured.

### Eye-opener for partners on the childbirth process

The lack of understanding among husbands about labor-related concerns was cited as one of the causes of disrespect and harassment at home. The presence of husbands in the delivery room was associated with a positive change in their attitudes toward women. Pregnant women believed that their husbands would be supportive in the event of an unexpected home birth if they witnessed the labor and delivery process. Additionally, having a partner present during birth enables the couple to receive health education together.

"*. . .being there for a partner to see how women suffer because there are some men who abuse their wives, sometimes they want sex immediately after being discharged from the hospital. . . we are the one who suffer for the children and we are considered as Automatic Teller*

*Machines (ATM) this happens because men don't knowledge about labor and delivery...*"
(FGD2, P1)

### Enhancement of women health

During labor and delivery, pregnant women require moral support from their partners. They also need someone to pray for them, massage their back, and help them change positions. Men are said to improve their attitudes toward women by witnessing the challenges of labor and delivery.

"*...the pain and hardships that I go through let him see. I believe, if he sees the process, he will be compassionate, and it will help in family planning...*" (FGD2 –P6)

According to healthcare providers, companionship motivates couples to use family planning, increases respect, and decreases embarrassment and abuse in the family.

"*...during that time of companionship, mother was gravida 12 (12th pregnancy) accompanied with her husband, after observing the process of childbirth; the husband requested for family planning to his wife...*" (IDI—06).

### Accountable partner during childbirth

Pregnant women noted that having a partner around would be beneficial since they could help with tasks like buying medications, giving oral sips, and massaging the mother. Healthcare providers said that the presence of a partner would reduce complaints and would positively represent the health facility in the community. Additionally, the partner would assist the laboring mother in making decisions that would result in the prompt management of labor.

"*...when a partner is there, it is easy to get requirements on time, is the one who is going to buy it and there will be no delay in getting services ...*" (FGD3 –P3)

### Assurance of safety to the mother

The presence of a companion, particularly the spouse or partner, was perceived as a way to ensure the safety of the mother during childbirth. Participants felt that having a familiar person present could help prevent medical errors and provide additional support in case of emergencies.

### Benefits of companion presence during labor

Due to the unfamiliar surroundings and faces in the labor ward, a pregnant woman's presence with her spouse or other companion throughout labor and delivery helps her feel secure. The likelihood of a prompt evaluation and a decrease in unnecessary cesarean sections were expressed as benefits. Additionally, both pregnant women and healthcare providers thought it was unlikely that a student would be allowed to examine the laboring woman if the companion were present.

"*...last year at one health facility in Kigoma, I lost my baby, the baby was very big, a nurse was yelling at me, why are you failing to push, she slapped me...later, another provider*

*assessed me, then decision for operation was made. . . I was operated but the baby died imme-diately due to exhaustion. If he (the husband) was there my baby wouldn't have died (crying. . .)" (FGD4-P4).*

Healthcare providers narrated on the importance of comfortable environment to the mother for better progress of labor. The presence of familiar face has positive effect on labor and delivery.

*". . .the progress of labor is good affected by four P's i.e. Power, passenger, passage and psy-che. . . calm environment due to the presence of trusted partner will decrease other interven-tions like use of oxytocin. . ." (IDI– 09)*

### Presence of emotionally attached person

Pregnant women preferred to have someone with whom they believed their information is safe during all phases of childbirth. Husbands were highly preferred because they are the parents of the newborn and have been involved since conception. Biological mothers were also favored due to their inherent knowledge of the delivery process and their propensity to protect a woman's privacy under any circumstances.

*". . .pregnancy is a plan of me and my husband, we are sleeping together and I won't be afraid of his presence because nothing is new to him. It is difficult to trust someone who is not famil-iar to me. I need my husband to be present during childbirth, psychologically will make me to feel comfortable because he is my secret partner. . ." (FGD1-P4)*

### Fear of blame and breach of confidentiality

Despite the benefits of companionship, both pregnant women and healthcare providers expressed fears related to privacy and confidentiality. Pregnant women were concerned about their health information being shared without consent, while healthcare providers worried about potential blame for medical errors if a companion was present.

### Accountability challenges in staffing and patient care

One aspect that reportedly contributed to the failure to implement companionship was a shortage of staff. In cases where a nurse did not assess a laboring mother timely, the existence of a companion may be helpful evidence to the partner if the mother and baby suffer harm. The presence of partners is thought to keep healthcare providers busy, ensuring that mistakes are not made.

*". . .in our hospitals, you may find one nurse or two saving more than two clients who needs regular assessments. . . You stuck here, you forgot there, what will happen? The partner has already noticed. . ." (IDI– 06)*

### Culture of covering gender violence

Healthcare providers were afraid to perform their duties in front of family members. They believed that when relatives are present, it is easier for them to identify their weaknesses than when they are working with a client alone. When a woman is uncooperative, a message is

occasionally conveyed in a harsh tone—unlikely to happen in the company of a relative. Due to their fear of being exposed for mistreatment, healthcare providers did not feel comfortable working in the presence of a companion. As a result, family members often end up waiting outside during care, despite their willingness to accompany the patient.

*"...mother may not listen to you at all...when you give service in the delivery room it needs to be a little harsh. I do not mean to beat the mother or use of abusive language, so when her partner is there, he can take it as if you have insulted or yelled at his patient..." (IDI– 03)*

## Society of fear

Most men are not prepared to witness the laboring mother and the delivery process. Financial difficulties have been linked to the worry that a family member would accompany a laboring mother, along with the responsibility of paying the bill and occasionally purchasing medicines. Some women fear being accompanied by their husbands because they believe their attractiveness to men will decrease as a result of watching the delivery process.

*"...with time experience has been shown that the society is not ready. Tanzanian men don't like to see a woman in delivering process due to fear of the process or cost associated issues, they don't want to be around his wife seeing her while laboring and delivering..." (IDI-10)*

## Lack of privacy and confidentiality

With a large number of women giving birth, the majority of labor and delivery rooms in public health institutions are inadequately designed for companionship. Privacy is provided by using curtains, which are inadequate and exposure is possible. Pregnant women complained that the delivery rooms did not offer complete privacy. Healthcare providers also noted that although the labor and delivery rooms had curtain dividers, their privacy is insufficient and uncomfortable for the partner.

*"...Delivering rooms are not friendly because the space is too small to accommodate companion comfortably and the curtain does not provide maximum privacy..." (IDI– 10)*

Overall, the findings suggest that while companionship during childbirth is highly valued, challenges related to privacy, confidentiality, and fear of blame need to be addressed to improve the practice.

## Discussion

Our study delved into the experiences of pregnant women and healthcare providers regarding childbirth companionship at Mbeya Zonal Referral Hospital. We gathered insights from both groups on their preferences and the barriers to childbirth companionship. Three key themes emerged: learning and bonding with the spouse/partner, assurance to the mother, and fear of blame and breach of confidentiality.

Both pregnant women and healthcare providers expressed a strong desire for companionship during childbirth at all stages of labor. Mothers valued having their husbands present during childbirth, as it allowed them to witness the process and understand the challenges women face during delivery. This desire for companionship aligns with findings from Ghana, where women believed that having a companion would provide emotional support and improve how

men treat their wives after witnessing childbirth [31]. Similarly, a study done in Kigoma found that both women and healthcare providers were very satisfied with having companions during all stages of childbirth, as companions assisted with the workload and provided emotional support [26]. However, not all women share this sentiment; some women feared that their husbands would lose sexual desire for them if they witnessed childbirth, a concern echoed in similar studies conducted in Kenya and Russia [3, 32].

In this study, husbands were the preferred companions during childbirth, with women believing that their presence would positively impact both the mother and the new baby. Mothers felt that having a close relative present would make them more comfortable, reduce the need for unnecessary cesarean sections, and ensure that services were provided by qualified personnel. This preference for husbands as companions was also observed in a study from Nigeria, where both women and their partners preferred male companions for their emotional support [18]. Mothers highlighted that having a companion, particularly their partner, could help avoid delays in care provision by assisting with non-professional tasks such as buying medicines. In some African cultures, husbands have the final say in family decisions, including healthcare choices, making their presence during childbirth crucial for timely decision-making [33].

Health workers in South Africa expressed discomfort working in the presence of companions, fearing their actions might be perceived as mistreatment [34]. This is similar to the findings in our study, where healthcare providers also felt uneasy with companions during childbirth. Such negative attitudes towards companionship may contribute to the underutilization of maternity services in government hospitals, leading to poor maternal and perinatal outcomes in both regions [33].

Several barriers to companionship during childbirth were identified by both healthcare providers and pregnant women. These barriers included a shortage of staff, fear of revealing mistakes, and concerns about breaching secrecy and confidentiality. Fear of blame has been identified as a major obstacle to quality improvement initiatives, such as implementing companionship during childbirth. In Kenya, providers prevented relatives from staying in the labor ward due to fear of mistakes being revealed if something went wrong, such as a baby dying during childbirth [19]. The design of the labor ward also posed challenges to implementing companionship, as it did not allow for the presence of a companion without compromising the privacy and confidentiality of other women in the room. This lack of privacy is a common issue in many public facilities in Tanzania and other low- and middle-income countries, as seen in Nigeria, where public-sector facilities often offer very little privacy due to design flaws and overcrowding [18].

While childbirth companionship is highly desired, its implementation in our setting is challenging due to facility, staff, personal, and sociocultural challenges. Health systems should strive to ensure that women have a positive experience in the delivery room, as having a companion is crucial for the psychological and financial well-being of the delivering woman. These strategies should address the identified barriers and leverage the benefits of companionship to improve maternal outcomes.

Addressing these barriers through targeted interventions and policy changes can enhance the childbirth experience for women in Tanzania, ensuring better psychological and physical outcomes. Future research should focus on long-term impacts and explore perspectives from a broader range of stakeholders to develop more inclusive and effective companionship practices.

## Conclusion

Despite this strong desire, the implementation of companionship faces numerous challenges, including facility design, staff shortages, and sociocultural barriers.

Healthcare providers acknowledge the benefits of companionship in improving maternal outcomes and reducing complaints about health services. Still, they also express concerns about revealing mistakes and discomfort working in the presence of companions, fearing their actions might be perceived as mistreatment.

Addressing these challenges requires a multifaceted approach. Health systems should prioritize the provision of adequate facilities that ensure privacy and confidentiality for laboring women and their companions. Staffing levels should be optimized to accommodate companions and ensure timely care provision. Additionally, efforts to raise awareness and change societal attitudes towards childbirth and companionship are crucial.

## Acknowledgments

We acknowledge MZRH administration and MUHAS IRB for permission and ethical clearance to conduct this research. Furthermore, we are thankful to MZH healthcare providers at maternity complex and pregnant women who attended ANC during the study period for their cooperation during data collection and anyone who made this research possible.

## Author Contributions

**Conceptualization:** Michael Obed Mwakyusa, Ali Said, Fadhlun M. Alwy Al-beity.

**Data curation:** Michael Obed Mwakyusa.

**Formal analysis:** Michael Obed Mwakyusa, Ali Said, Shekha Selemani, Judica Christopher, Nathanael Shauri Sirili.

**Methodology:** Michael Obed Mwakyusa, Shekha Selemani.

**Supervision:** Ali Said.

**Writing – original draft:** Michael Obed Mwakyusa, Shekha Selemani, Musa Kakiziba.

**Writing – review & editing:** Ali Said, Nathanael Shauri Sirili, Fadhlun M. Alwy Al-beity.

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
