## [Decision Letter · Decision Letter 0]

19 Oct 2023

PONE-D-23-19544“If my husband was in the labor ward with me, my baby wouldn’t have died”; Experiences on birth companionship from a tertiary heath facility in southern highland, TanzaniaPLOS ONE

Dear Dr. Mwakyusa,

Thank you for submitting your manuscript to PLOS ONE. After careful consideration, we feel that it has merit but does not fully meet PLOS ONE’s publication criteria as it currently stands. Therefore, we invite you to submit a revised version of the manuscript that addresses the points raised during the review process.

We look forward to receiving your revised manuscript.

Kind regards,

Ivan Sarmiento

Academic Editor

PLOS ONE

Journal Requirements:

3. We note you have included a table to which you do not refer in the text of your manuscript. Please ensure that you refer to Table 1 and 2 in your text; if accepted, production will need this reference to link the reader to the Table.

Reviewers' comments:

Reviewer's Responses to Questions

**Comments to the Author**

1. Is the manuscript technically sound, and do the data support the conclusions?

Reviewer #1: No

Reviewer #2: Yes

2. Has the statistical analysis been performed appropriately and rigorously? 

Reviewer #1: N/A

Reviewer #2: N/A

3. Have the authors made all data underlying the findings in their manuscript fully available?

Reviewer #1: Yes

Reviewer #2: No

4. Is the manuscript presented in an intelligible fashion and written in standard English?

Reviewer #1: No

Reviewer #2: No

5. Review Comments to the Author

Reviewer #1: Summary of reviewer comments:

The comments below highlight several areas for improvement in the research paper. Notably, there are suggestions to enhance the clarity and specificity of the title, abstract, and introduction, emphasizing the importance of clearly defining the problem and its context. The methodology section needs revisions for better clarity, particularly in specifying the type of data collection used. The study setting details are deemed excessive, and the reviewer recommends focusing on essential information. Spelling and grammar issues throughout the document need addressing. In the results section, there are concerns about the placement and necessity of certain tables, as well as the need for more quotes and citations to support themes. The discussion section requires a more organized structure, critical analysis, and thorough exploration of previous research in Tanzania. Additionally, the reviewer suggests addressing limitations, providing practical implications, and discussing potential policy implications. Overall, there is a call for increased coherence, consistency, and attention to detail throughout the paper.

Discussion of specific areas for improvement:

- Introduction Clarity: The introduction requires restructuring to better articulate the problem from the outset. Beginning with a clearer delineation of the research problem, whether it be focused on respectful maternity care or broader maternal and fetal/neonatal outcomes, will provide readers with a more concise and engaging introduction.

- Methodology Section: The methodology section demands a thorough revision. The first sentence needs to be rewritten for better clarity and smoother sentence structure. Additionally, the type of data collection (interviews or focus groups) should be explicitly stated. The excessively detailed description of the study setting, including GPS coordinates and unrelated hospital information, should be streamlined for conciseness.

- Spelling and Grammar: A comprehensive review for spelling and grammar issues throughout the document is crucial. Instances such as the capitalized "focused" and unexplained acronyms like "FGD" and "IDIs" need correction for linguistic precision.

- Results Presentation: The placement and necessity of tables in the results section need justification. It is advisable to either integrate the information into the text or provide a brief explanation for the tables. Furthermore, there is a need for additional quotes and citations to substantiate the identified themes and sub-themes.

- Discussion Section Organization: The discussion section requires a more organized structure, including a clearer reintroduction and elaboration of the themes identified in the results. The reviewer also recommends a more critical analysis of the results, addressing limitations, and providing practical and policy implications. Additionally, exploring existing research in Tanzania and its relevance to the current study will enhance the depth of the discussion.

The manuscript has some major technical and methodological issues that need to be addressed to ensure it meets the criteria of being technically sound. The comments highlight concerns about clarity in the title, abstract, and introduction, as well as issues related to methodology, data presentation, and the organization of the discussion. Some key areas for improvement include refining the title to clearly convey the methodology, restructuring the introduction for better problem definition, addressing issues in the methodology section (e.g., clarity on data collection type, excessive study setting details), and improving the organization and critical analysis in the discussion section. These suggestions collectively indicate that revisions are needed to enhance the technical soundness and rigor of the manuscript.

Reviewer #2: Authors present a pertinent study for women healthcare in Tanzania. It is interesting to have women voices as well as health care professionals, even more when they both conclude the need of strengthening companionship during labor in the study setting.

Denial of companionship in labor is a manifestation of obstetric violence that enhances the possibility of other mistreatments during labor (e.g. or e.g.2). This is important to address in introduction, results and discussion as there are important results that indicate obstetric violence lived by participants in the study setting.

Revisions in each section of the report are described in the Attachment.

6. PLOS authors have the option to publish the peer review history of their article (what does this mean?). If published, this will include your full peer review and any attached files.

Reviewer #1: No

Reviewer #2: No

---

## [Author Response · Author response to Decision Letter 0]

16 Apr 2024

Comments from both reviewers has been addressed accordingly.

---

## [Editor Report · Decision Letter 1]

29 Apr 2024

PONE-D-23-19544R1“If my husband was in the labor ward with me, my baby wouldn’t have died”; Experiences on birth companionship from a tertiary heath facility in TanzaniaPLOS ONE

Dear Dr. Mwakyusa,

Thank you for submitting your manuscript to PLOS ONE. After careful consideration, we feel that it has merit but does not fully meet PLOS ONE’s publication criteria as it currently stands. Therefore, we invite you to submit a revised version of the manuscript that addresses the points raised during the review process.

Having a proper rebuttal letter with responses to each of the reviewers' comments is essential. I am resending the comments that we received from the peer review. 

Reviewere#1

Summary of reviewer comments:

The comments below highlight several areas for improvement in the research paper. Notably, there are suggestions to enhance the clarity and specificity of the title, abstract, and introduction, emphasizing the importance of clearly defining the problem and its context. The methodology section needs revisions for better clarity, particularly in specifying the type of data collection used. The study setting details are deemed excessive, and the reviewer recommends focusing on essential information. Spelling and grammar issues throughout the document need addressing. In the results section, there are concerns about the placement and necessity of certain tables, as well as the need for more quotes and citations to support themes. The discussion section requires a more organized structure, critical analysis, and thorough exploration of previous research in Tanzania. Additionally, the reviewer suggests addressing limitations, providing practical implications, and discussing potential policy implications. Overall, there is a call for increased coherence, consistency, and attention to detail throughout the paper.

Discussion of specific areas for improvement:

- Introduction Clarity: The introduction requires restructuring to better articulate the problem from the outset. Beginning with a clearer delineation of the research problem, whether it be focused on respectful maternity care or broader maternal and fetal/neonatal outcomes, will provide readers with a more concise and engaging introduction.

- Methodology Section: The methodology section demands a thorough revision. The first sentence needs to be rewritten for better clarity and smoother sentence structure. Additionally, the type of data collection (interviews or focus groups) should be explicitly stated. The excessively detailed description of the study setting, including GPS coordinates and unrelated hospital information, should be streamlined for conciseness.

- Spelling and Grammar: A comprehensive review for spelling and grammar issues throughout the document is crucial. Instances such as the capitalized "focused" and unexplained acronyms like "FGD" and "IDIs" need correction for linguistic precision.

- Results Presentation: The placement and necessity of tables in the results section need justification. It is advisable to either integrate the information into the text or provide a brief explanation for the tables. Furthermore, there is a need for additional quotes and citations to substantiate the identified themes and sub-themes.

- Discussion Section Organization: The discussion section requires a more organized structure, including a clearer reintroduction and elaboration of the themes identified in the results. The reviewer also recommends a more critical analysis of the results, addressing limitations, and providing practical and policy implications. Additionally, exploring existing research in Tanzania and its relevance to the current study will enhance the depth of the discussion.

The manuscript has some major technical and methodological issues that need to be addressed to ensure it meets the criteria of being technically sound. The comments highlight concerns about clarity in the title, abstract, and introduction, as well as issues related to methodology, data presentation, and the organization of the discussion. Some key areas for improvement include refining the title to clearly convey the methodology, restructuring the introduction for better problem definition, addressing issues in the methodology section (e.g., clarity on data collection type, excessive study setting details), and improving the organization and critical analysis in the discussion section. These suggestions collectively indicate that revisions are needed to enhance the technical soundness and rigor of the manuscript.

Summary of reviewer comments:

The comments below highlight several areas for improvement in the research paper. Notably, there are suggestions to enhance the clarity and specificity of the title, abstract, and introduction, emphasizing the importance of clearly defining the problem and its context. The methodology section needs revisions for better clarity, particularly in specifying the type of data collection used. The study setting details are deemed excessive, and the reviewer recommends focusing on essential information. Spelling and grammar issues throughout the document need addressing. In the results section, there are concerns about the placement and necessity of certain tables, as well as the need for more quotes and citations to support themes. The discussion section requires a more organized structure, critical analysis, and thorough exploration of previous research in Tanzania. Additionally, the reviewer suggests addressing limitations, providing practical implications, and discussing potential policy implications. Overall, there is a call for increased coherence, consistency, and attention to detail throughout the paper.

Discussion of specific areas for improvement:

Introduction Clarity: The introduction requires restructuring to better articulate the problem from the outset. Beginning with a clearer delineation of the research problem, whether it be focused on respectful maternity care or broader maternal and fetal/neonatal outcomes, will provide readers with a more concise and engaging introduction.Methodology Section: The methodology section demands a thorough revision. The first sentence needs to be rewritten for better clarity and smoother sentence structure. Additionally, the type of data collection (interviews or focus groups) should be explicitly stated. The excessively detailed description of the study setting, including GPS coordinates and unrelated hospital information, should be streamlined for conciseness.Spelling and Grammar: A comprehensive review for spelling and grammar issues throughout the document is crucial. Instances such as the capitalized "focused" and unexplained acronyms like "FGD" and "IDIs" need correction for linguistic precision.Results Presentation: The placement and necessity of tables in the results section need justification. It is advisable to either integrate the information into the text or provide a brief explanation for the tables. Furthermore, there is a need for additional quotes and citations to substantiate the identified themes and sub-themes.Discussion Section Organization: The discussion section requires a more organized structure, including a clearer reintroduction and elaboration of the themes identified in the results. The reviewer also recommends a more critical analysis of the results, addressing limitations, and providing practical and policy implications. Additionally, exploring existing research in Tanzania and its relevance to the current study will enhance the depth of the discussion.

More specific comments:

Title and Abstract: Consider revising the title to clearly indicate the study's methodology. Additionally, replace "Southern Highland" with descriptors like "rural" or "urban" for better reader comprehension.Abstract background: The abstract's background requires reworking to immediately engage the reader. Start by clearly defining the problem—whether it's focused on respectful maternity care or broader maternal and fetal/neonatal outcomes. Introduce why studying birth companionship is crucial in addressing these outcomesAbstract objective: Reconsider using "analyze" in the abstract objective. In qualitative studies, terms like "explore" or "examine" would align better with the research approach.Abstract methodology: Rewrite the first sentence of the abstract methodology for improved clarity and smoother sentence structure. Consider: "A qualitative exploratory approach was utilized for interviews with purposefully selected participants from September to October 2021."Abstract methodology: Clarify the type of data collection used (interviews or focus groups) for better understanding.Conduct a thorough check for spelling and grammar errors. Address the unnecessary capitalization of "focused" and clarify terms like "antenatal clinic."Abstract results: Avoid starting sentences with "But" for grammatical correctness. Revise sentence structure and spelling for improved readability.Introduction: Consider starting the introduction with paragraph 2, focusing on defining the problem from the outset. Major attention is needed for spelling and grammar throughout this section.Review references in paragraph 1. Avoid using "WHO recommends..." without referencing specific articles or texts from the WHO. Reframe sentences to refer to previous research.Introduction, paragraph 3. Ensure references are appropriately placed throughout. Provide clarity on the definition of labor and expand on factors influencing it.Introduction paragraph 5: Elaborate on how companionship can be influenced by various factors. Provide examples, such as age or religious background.Clarify the importance of healthcare providers' perceptions. Consider including a section on previous research detailing healthcare providers' views on companionship, especially in LMICs or sub-Saharan Africa.Introduction: Avoid unnecessary repetition of the WHO abbreviation in the introduction.Introduction: Clearly state the motivation behind the project. Specify the gap in previous research in Tanzania or sub-Saharan Africa concerning birth companionship and its effects on maternal outcomes. Where is the why? Why are you doing this project?Again reword your objective. Revise the objective to better align with the motivation for exploring the experiences of both pregnant women and healthcare providers. Clarify why both perspectives are essential and how they contribute to addressing potential miscommunication.Methods: Avoid reannouncing the aim in the methods section. Ensure adherence to PLOS instructions for the methods structure.Methods, study design: Justify the choice of an exploratory qualitative study and its alignment with the research objective. Provide information about the specific area in Tanzania.Methods, Study setting: Streamline the study setting section by omitting unnecessary details. Focus on essential information, such as the location in the south of Tanzania, and provide a clear rationale for choosing this zone.Methods: a map of the study setting could be useful instead of all of this text which is entirely unnecessary.Method, study setting: Trim down information on the hospital and focus on critical details, such as the number of women in the maternity ward, annual deliveries, and maternal outcomes.Method, study setting: Reposition the study setting objective, placing it more logically as the overall study objective, as it succinctly describes the research focus in Tanzania. “Therefore, this study aimed to explore desire, preferred childbirth companionship and perceived barriers among pregnant women and healthcare providers in Tanzania.”Methods, participants and recruitment: Streamline the participants and recruitment section. Clearly identify the roles of co-authors and provide concise details on the sampling strategies and reasons for participant recruitment.Methods: Provide clarity on the decision-making process regarding the exclusion of focus groups with healthcare providers.Methods: Condense the methods section, considering the inclusion of supplementary material or an appendix for excessive details.Method, data collection and analysis: again specify why co author did what for the in depth interviews.Method, data collection and analysis: Specify the roles of co-authors in in-depth interviews and focus groups. Describe how the interview guide for the interviews and focus groups was developed, whether within the research team or based on a framework. Justify and provide references for the in-depth interview and semi-structured guide.Method: Define acronyms like FGD and IDIs to enhance reader understanding.Method: Clarify the type of coding employed in the research—whether it was inductive or deductive.Results: Justify the placement of tables in the results section and ensure they are appropriately described or integrated into the text. Clearly indicate whether the themes apply to both women and healthcare providers.Results: Evaluate the necessity of Table 2 and, if retained, ensure it is justified and referenced appropriately.Results: Strengthen themes and subthemes with more quotes and citations to substantiate findings. Clarify the connection between subthemes and the overall theme. Are these themes for both the women and the healthcare providers. Very unclear. It almost seems like these could be two different studies? One looking at womens opinion and one at healthcare providers? Why combine? Please provide reasoning and justification? Some of themes or sub themes need more quotes or citations to back up the sub themes. As well it is unclear how the sub themes are connected to the overall theme that you described?Discussion: Revisit the first paragraph of the discussion to reintroduce and describe the themes again.Discussion (Previous Research in Tanzania): Enhance the discussion by incorporating more information about previous research in Tanzania, both qualitative and quantitative, on birth companionship and its impact on maternal outcomes.Discussion (Organization): Improve the organization of the discussion section by grouping related findings together for better flow and readability.Discussion (Citations): Verify the presence and appropriateness of citations in the discussion section.Discussion (Redundancy): Address redundancy in discussing the integration of study results with those of previous research.Discussion (Practical Implications): Include more discussion on the practical implications of the research and how the results contribute to the existing body of knowledge. Highlight what is novel in this research.Discussion (Critical Analysis): Move beyond summarizing findings and provide a more critical analysis of the results.Discussion (Limitations): Introduce a discussion of limitations and potential biases within the study.Discussion (Barriers to Companionship): Delve deeper into the barriers to implementing childbirth companionship. Provide detailed insights into challenges at both institutional and cultural levels.Discussion (Future Research): Conclude the discussion with suggestions for future research, emphasizing existing gaps in knowledge and proposing ways to address them.Discussion (Consistency): Ensure consistency in language and style throughout the discussion. Avoid interchanging terms like "companionship" and "companion."Discussion (Policy Implications): Discuss potential policy implications based on the findings. Outline how healthcare policies could be shaped or modified to accommodate preferences and needs identified in the study.

Reviewer #2

Authors present a pertinent study for women healthcare in Tanzania. It is interesting to have women voices as well as health care professionals, even more when they both conclude the need of strengthening companionship during labor in the study setting.

Mayor revision

Denial of companionship in labor is  a manifestation of obstetric violence that enhances the possibility of other mistreatments during labor (e.g. or e.g.2). This is important to address in introduction, results and discussion as there are important results that indicate obstetric violence lived by participants in the study setting.

Revisions in each section of the report:

**Abstract**

Simplify introduction: Consider to start the paragraph with the third phrase and to merge two first sentences.  Include a phrase resuming the justification of the study in the context of Tanzania (e.g.  even though there are politics to enhance companionship, it is not being very common).

Key words: consider to include standardized terms (e.g. maternal healthcare) and exclude “desire” as it is unspecific to the study

**Introduction**

The authors develop a precise overview of childbirth companionship and maternal healthcare. A focus on local context on the matter needs to be given in introduction (maternal healthcare and companionship in Tanzania; socio economic differences between hospitals, cultural / religious context) that may influence the phenomenon.p. 4. Interesting evidence around preferences of women companionships, influenced by education and beliefs.  Please describe examples of these factors.

**Methodology**

**Study setting **

Please clarify socioeconomic conditions of patients attended in the selected hospital.Revise the last phrase that presents the purpose of the study. Consider to correct by excluding the word “desire”.

**Participants:**

Good rationale related to sampling. Revise writing to gain clarity in this procedure.The first time to include an acronym, also include the words being summarized.It is necessary to explain sampling/recruitment procedure with the mothers` group.Include a reference to Table 1.

**Data collection:**

Please explain if in the focus groups design authors considered any segmentation criteria to group participants.

**Analysis:**

Please separate this section from data collection.

**Trustworthiness:**

Please clarify the acronyms IDI and FGD.  If a maximum variation sampling was performed, it should also be described in the participants section along with the chain strategy described.As Authors developed initial coding individually, you can also declare triangulation as a trustworthiness strategy.There is a need of positionality and reflexivity processes declarations of the authors as physicians. ¿Do authors work in the institution?  this is also important to be transparent with possible interests in conflict in the study. ¿Are participants also their patients?

**Results**:

Table 1. Please clarify what does Standard 7 (education) for international audience. what are the health professions included in the interviews?The paragraph below table 2 can be removed as it presents the same information of the table.  Authors may consider instead a brief resume of the particular differences on the results between the two groups.The theme “creation of close relationships in the family” seems to be related to the enhancement of women health, by reducing gender violence and inequity from the perspective of women and health professionals.  This is an important result that deserves to be highlighted.Theme “mother having a sense of security”: The strong institutional and interpersonal violence experienced by the woman participant during labor needs to me addressed by authors in their description of the theme. The four “Ps” described in results seems to be part of the discussion. Something similar is found in “society of fear”. Remember to focus on results expressed by participants in this section.Theme: “Staffs are avoiding accountability for their shortcomings”: It is important to clarify this theme and revise the quote writing and punctuation to gain clarity. Is this something said only by the staff?Theme “culture of covering mistakes”:  The authors description and participant quote evidence a culture of covering gender violence, particularly during labor. Not simple “mistakes”.  evidence reported in previous theme also showed gender violence in women healthcare service during labor. This is highly important and should be evident in the name of the theme: it enhances the relevance of their companionship, and shows institutional and structural barriers to the phenomenon under study.

**Discussion and conclusion:**

This section can be strengthened framing the results in the global aim of a respected childbirth healthcare. The consulted WHO guidelines to enhance childbirth companionships can also guide discussion related to the identified barriers.Results related to obstetric violence should be discussed in this section. Please frame conclusion in the context and scope of the study.

**Data availability:  **please clarify where or how is the data available.

**Review English writing:  **

Please review English writing. There are multiple semicolons that might not be appropriate in the paragraph.Introduction: first paragraph (The who recommends…; However, in real clinical situations). Also review wording of third paragraph, first phrase.p.6. “Mbeya zonal referral hospital has obstetrics and gynecology department which saves the people from different parts of Mbeya city and Southern highland zone with complicated obstetric and gynecological conditions” (Do author mean “attends”?)p. 7. “and focused group discussion to the with pregnant women”

We look forward to receiving your revised manuscript.

Kind regards,

Ivan Sarmiento

Academic Editor

PLOS ONE

---

## [Author Response · Author response to Decision Letter 1]

4 Jun 2024

Matrix table of comments and responses attached

---

## [Decision Letter · Decision Letter 2]

2 Jul 2024

PONE-D-23-19544R2“If my husband was in the labor ward with me, my baby wouldn’t have died”; Experiences on birth companionship from a tertiary heath facility in TanzaniaPLOS ONE

Dear Dr. Mwakyusa,

Thank you for submitting your manuscript to PLOS ONE. After careful consideration, we feel that it has merit but does not fully meet PLOS ONE’s publication criteria as it currently stands. Therefore, we invite you to submit a revised version of the manuscript that addresses the points raised during the review process.

Thank you for your efforts in enhancing the manuscript. Please review the additional comments from a reviewer and the detailed list of comments in the attached document.

We look forward to receiving your revised manuscript.

Kind regards,

Ivan Sarmiento

Academic Editor

PLOS ONE

**Additional Editor Comments:**

Please see the attached document for detailed comments on the manuscript.

Reviewers' comments:

Reviewer's Responses to Questions

**Comments to the Author**

1. If the authors have adequately addressed your comments raised in a previous round of review and you feel that this manuscript is now acceptable for publication, you may indicate that here to bypass the “Comments to the Author” section, enter your conflict of interest statement in the “Confidential to Editor” section, and submit your "Accept" recommendation.

Reviewer #1: (No Response)

2. Is the manuscript technically sound, and do the data support the conclusions?

Reviewer #1: Partly

3. Has the statistical analysis been performed appropriately and rigorously? 

Reviewer #1: N/A

4. Have the authors made all data underlying the findings in their manuscript fully available?

Reviewer #1: Yes

5. Is the manuscript presented in an intelligible fashion and written in standard English?

Reviewer #1: No

6. Review Comments to the Author

Reviewer #1: Hello,

Thank you for the significant improvements and for thoroughly addressing the comments. Here are my additional suggestions that still warrant a minor revision:

Abstract:

Include the number of participants in the abstract, specifying how many women were selected for the focus groups and how many providers were interviewed.

Introduction:

Merge paragraphs 6 and 7 into one paragraph since both discuss providers and companionship.

Once you introduce the abbreviation of the WHO, ensure you consistently use this abbreviation throughout the introduction. If not, avoid introducing the abbreviation.

The last paragraph lacks important references regarding the context of Tanzania. It would be beneficial to include citations of previous studies on this subject in Tanzania.

Methods:

References are missing about the number of deliveries in the hospital each month. It's crucial to back up your statements with data.

There is no information about how many women were recruited and the approximate number of women in each focus group. Please provide this range.

Include information on how many months pregnant the women were. This detail is important when discussing the companionship experience. Additionally, explain the decision to recruit pregnant women instead of postpartum women.

Results:

At the start of each theme, include introductory sentences summarizing the sub-themes and explaining how they relate to the overall theme. This will address the current disconnect and make the results section more cohesive and narrative.

Discussion:

The first paragraph of the discussion needs another sentence to describe the themes in more detail and provide a more comprehensive summary of the results.

Overall, there are still some sentence structure issues, referencing mistakes, and paragraph structuring problems that need to be addressed.

7. PLOS authors have the option to publish the peer review history of their article (what does this mean?). If published, this will include your full peer review and any attached files.

Reviewer #1: No

---

## [Author Response · Author response to Decision Letter 2]

29 Jul 2024

The document of Responses to reviewer has been attached

---

## [Editor Report · Decision Letter 3]

15 Aug 2024

“If my husband was in the labor ward with me, my baby wouldn’t have died”; Experiences on birth companionship from a tertiary health facility in Tanzania

PONE-D-23-19544R3

Dear Dr. Mwakyusa,

We’re pleased to inform you that your manuscript has been judged scientifically suitable for publication and will be formally accepted for publication once it meets all outstanding technical requirements.

Kind regards,

Ivan Sarmiento

Academic Editor

PLOS ONE
---

## [Editor Report · Acceptance letter]

19 Aug 2024

PONE-D-23-19544R3 

PLOS ONE

Dear Dr. Mwakyusa, 

I'm pleased to inform you that your manuscript has been deemed suitable for publication in PLOS ONE. Congratulations! Your manuscript is now being handed over to our production team.

Kind regards, 

on behalf of

Dr. Ivan Sarmiento 

Academic Editor

PLOS ONE